# Lived experience of cognitive-communication changes for people with acquired brain injury and familiar communication partners: A qualitative evidence synthesis

Nicholas Behn[1]*, Iben Christensen[2], Madeline Cruice[1], Katerina Hilari[1], Ian Kellar[3], Leanne Togher[4]

1 School of Health and Medical Sciences, City St George's, University of London, United Kingdom,
2 Department of Nordic Studies and Linguistics, University of Copenhagen, Denmark, 3 School of Psychology, The University of Sheffield, United Kingdom, 4 Discipline of Speech Pathology, Faculty of Medicine and Health, The University of Sydney, Australia

* Nicholas.behn@citystgeorges.ac.uk

## Abstract

### Background and objectives

Cognitive-communication disorder (CCD) is common after acquired brain injury (ABI), reported in about two-thirds of people who sustain an injury. Quantitative studies have found that the disorder can negatively impact a person's ability to socially re-integrate into the community, return to work or education and achieve a good quality of life. However, little is known about *how* the disorder impacts people with ABI and the family. Therefore, the aim of this qualitative evidence synthesis was to provide a detailed exploration of the lived experience of CCD for people with ABI and their family members.

### Methods

A systematic literature search was conducted across eight databases (CINAHL Ultimate, PsycINFO, PsycARTICLES, Medline, EMBASE, AMED, Scopus, PubMed) to August 2025. Studies were included if they reported on people with ABI who present with CCD (or similar term) and/or familiar communication partners whereby the impact of the disorder was described. Relevant data were extracted, and studies were critically appraised using the Critical Appraisal Skills Programme (CASP) qualitative checklist and the confidence of the findings was assessed using GRADE-CERQual tool. The final included studies were synthesised using thematic analysis.

### Results

13 articles met the eligibility criteria and reported on 103 people with ABI with CCD and 66 familiar communication partners including spouses, parents, friends, carers,

**Data availability statement:** All relevant data are within the manuscript and its Supporting Information files.

**Funding:** This study is funded by the NIHR Advanced Fellowship (NIHR302952). The views expressed are those of the author(s) and not necessarily those of the NIHR or the Department of Health and Social Care. The funders had no role in study design, data collection and analysis, decision to publish, or preparation of the manuscript.

**Competing interests:** No authors have competing interests.

siblings and children. Methodologies comprised interviews (n = 10), focus groups (n = 1), spoken discourse samples (n = 1) and online survey (n = 1). Eight main analytic themes were identified centred around the experiences of both people with ABI: (1) communicating is not easy; (2) lack of awareness and feeling tired; (3) anxiety, embarrassment and isolation; (4) connecting with others; and (5) participation and identity; and their familiar communication partner: (6) adjusting to giving increased support; (7) emotional toll of supporting; (8) relationship and life role changes.

## Conclusions

This review highlights the broad and unique impacts of CCD for both people with ABI and their familiar communication partners. People with ABI require tolerance to manage their communication difficulties; and communication partners require education, support and training to manage the change in relationship. These findings underpin the need for interventions to include partners in rehabilitation and for therapists to consider the diverse needs of people with ABI including emotions, relationships, social participation and changes to identity.

## Introduction

Acquired brain injury (ABI) is a leading cause of long-term disability worldwide, associated with substantial social and economic costs. In the United Kingdom alone, ABI contributes to an estimated £43 billion in annual healthcare and societal expenditure [1], with over 1.3 million people living with the long-term consequences of brain injury and approximately 350,000 hospital admissions each year [2]. Globally, the burden of brain injury has been estimated at £282 billion annually [3]. ABI encompasses both traumatic causes (e.g., road traffic accidents, falls, assaults) and non-traumatic causes (e.g., stroke, tumours, anoxia, encephalitis). Across these aetiologies, communication difficulties, also referred to as cognitive-communication disorders (CCD) are common for more than two-thirds of individuals following injury [4–6].

CCD is a multifaceted impairment arising from disruptions to cognitive processes that underpin communication, including attention, memory, executive functioning, and social cognition [7]. As a result, communication changes are heterogeneous, encompassing verbosity or paucity of speech, difficulties with initiation and turn-taking, tangential discourse, perseveration, impaired topic management, disinhibited behaviours, and breakdowns in pragmatic use of language [8–10]. These impairments rarely occur in isolation, and are often accompanied by emotional, behavioural, and physical changes, making rehabilitation complex [11]. Furthermore, these changes can be influenced by premorbid variables including literacy proficiency, gender identity and cultural linguistic background [12].

The consequences of CCD are profound with reported correlations between deleterious communication changes post-injury and reduced participation and/or poorer psychosocial outcomes. Changes in communication can hinder social integration

[13–16], limit opportunities for return to work or education [16–18] and reduce quality of life [13,19]. However, these studies provide limited information about how exactly these changes influence an individual with ABI and familiar communication partners, who may include spouses, parents, siblings, friends and/or carers.

Where studies have explored the lived experience of brain injury, communication changes are shown to be an influential factor even when not the primary focus of investigation, which underscores their pervasive impact on a person's life post-injury. Communication changes may lead to emotional distress, low self-confidence, social isolation, relationship breakdown, disrupt new romantic relationships and challenge a return to work [20–24]. In interviews and focus groups involving 62 people with ABI, Schipper and colleagues [25] found that a person's communication skills and interactions with others may negatively influence social participation including difficulties in expressing needs and wishes to others. In a qualitative study of 11 people with traumatic brain injury, Salas and colleagues [26] found that changes to communication including difficulty formulating thoughts and getting messages across clearly may negatively impact a person's social interactions, leading to frustration and isolation. Moreover, family members report feelings of burden, frustration, and loss, with communication changes leading to strained relationships and reduced family functioning [27].

Many studies include mixed aetiologies including both stroke and traumatic brain injury, with different communication diagnoses including CCD. In some studies, the communication diagnoses are not reported for participants [21–23,26] making it difficult to determine the type of communication change that has occurred. Other studies refer to the presence of aphasia [20,25] or other speech- or language-related communication diagnoses, such as dysarthria or apraxia of speech, which are quite distinct from cognitive-communication changes.

There is an inherent need for a deeper understanding of the lived experience of cognitive-communication changes. Such an understanding may help inform person-centred rehabilitation and ensure treatment is relevant to the real-life challenges individuals are experiencing post-injury. Therefore, the aim of this research was to review and explore the existing research *cognitive-communication changes* and the lived experience for people with ABI and familiar communication partners including, how such changes are managed and negotiated in daily life.

## Methods

To understand the lived experiences of cognitive communication changes for people with ABI and their communication partners, a qualitative evidence synthesis was undertaken [28]. Thematic synthesis was selected as the most appropriate method for this study [29] as it allows exploration of the participant experience. This method is flexible as to the type of data from primary research that can be synthesised, allowing both "thin" and "thick" data to be incorporated in the development of analytical themes [28]. The present review protocol is registered with PROSPERO (CRD42024519686). The PRISMA statement [30] and ENTREQ checklist [31] were used to ensure transparency in reporting the synthesis of qualitative research (see S1 and S2 File). Ethics approval was not required for this review given that all data used were collected from studies that are publicly available.

### Search strategy

Preliminary searches were conducted in selected databases to refine relevant keywords and search terms using the SPIDER (Sample-Phenomenon of Interest-Design-Evaluation-Research type) method for qualitative systematic searches [32,33]. Terms covering qualitative research were drawn from those developed by Barroso and colleagues [34]. The complete set of search terms is shown in Table 1. Search terms were entered into eight electronic bibliographic databases. Searches were conducted through EBSCOhost incorporating CINAHL Ultimate, APA PsycINFO, APA PsycArticles, Medline databases; and OVID incorporating Embase and AMED together with Scopus, and PubMed. The initial search was conducted in May 2024 and rerun in August 2025 using the limiters of human and adult (18+). There were no restrictions on language or publication date.

**Table 1. Database search terms.**

| SPIDER tool[a] | Search terms |
|---|---|
| S – Sample (title/abstract) | "traumatic brain injur*" OR "TBI" OR "ABI" OR "head injur*" OR "brain damage*" OR "head trauma" OR "brain injur*" OR "craniocerebral trauma*" OR "meningitis" OR "encephal*" OR "right hemisphere stroke" OR "right-hemisphere stroke" OR "arteriovenous malformation" OR "aneurysm" OR "brain haemorrhage" OR "cerebral haemorrhage" OR "brain tumo*" OR "cerebral tumo*" OR "brain neoplasm" OR "neurosurgery" OR "hypoxi*" OR "brain cancer" OR "glio*" OR "intracranial tumo*" or "intracranial neoplasm". |
| P of I – Phenomenon of interest (title/ abstract) | "Communication changes" OR "Communication disorder*" OR "communication dysfunction" OR "communication disability*" OR "communication problem*" OR "communication impairment*" OR "Communicative disorder*" OR "communicative dysfunction" OR "communicative disability*" OR "communicative problem*" OR "communicative impairment*" OR "cognitive communicati*" OR "cognitive-communicati*" OR "cognitive/communicati*" OR "cognitive and/or communicati*" OR "cognitive linguistic" OR "cognitive-linguistic" OR "cognitive language" OR "cognitive-pragmatic" OR "cognitive pragmatic" OR "high level language" OR "high-level language" OR "higher level language" OR "higher order language" OR "right hemisphere language" OR "right-hemisphere language" OR "pragmatic*" OR "discourse" OR "social cogniti*" OR "social perception" OR "theory of mind" OR "social-communication" OR "social communication" OR "social (pragmatic)" OR "acquired communication*" OR "acquired language*" OR "sub-clinical aphasia" OR "subclinical aphasia" |
| D – Design (all text) | "Case study" OR "constant compar*" OR "content analysis" OR "conversation analysis" OR "descriptive study" OR "discourse analysis" OR "exploratory study" OR "focus group" OR "grounded theory" OR "hermeneutic" OR "interview" OR "semi-structured" OR "narrative analysis" OR "ethnograph*" OR "naturalistic study" OR "participant observation" OR "phenomenolog*" OR "thematic analysis" OR "interpretative" OR "personal construct theory" OR "psychoanaly*" OR "framework analysis" OR "acceptability" OR "survey" OR "questionnaire" |
| E – Evaluation (all text) | "view*" OR "experienc*" OR "opinion*" OR "attitude*" OR "percep*" OR "perspective*" OR "belie*" OR "feel*" OR "know*" OR "understand*" OR "acceptability" OR "patient satisfaction" OR "satisf*" OR "value*" |
| R – Research type (all text) | "qualitative" OR "mixed method*" OR "mixed design" OR "mixed-design" OR "mixed-method*" |

[a][S AND P of I] AND [(D or E) AND R].

### Study selection

Search results from the nine databases were combined into Rayyan software [35] for study selection, with duplicates removed using Deduplicator [36]. Initially, 1000 random unique records from the wider search were imported to Rayyan, and the inclusion and exclusion decisions on this training set were then used to train Rayyan's machine learning prioritization feature (an SVM classifier)(i.e., "compute ratings" feature) to recognise and prioritise the most relevant records through finding patterns and similarities in articles included or excluded [37]. This function models the probability of inclusion for the remaining records by finding patterns in the titles and abstracts of the screened articles [37]. It then assigns each unscreened record a relevance rating, allowing the review to proceed by screening the most relevant articles first according to one of five labels: most likely to exclude (double thumbs down), likely to exclude (single thumbs down), no recommendation (question mark), likely to include (single thumbs up), and most likely to include (double thumbs up). This step was completed for both reliability purposes and to explore the utility of using text mining algorithms in the completion of systematic reviews. The titles and abstracts of 1000 random unique records were screened independently by two

reviewers (NB and IC) for inclusion for full-text review. Authors had a moderate level of agreement (k = 0.60; Landis and Koch, 1977 [38] and 97% agreement, with disagreements resolved through consensus. All remaining unique records were then imported for review by the first author with the text mining algorithm used again to automatically prioritise the most relevant records. A full text review of all included articles was completed by the two reviewers (NB and IC). Additional studies were identified through reference list checks of systematic reviews and checked for inclusion in the review. Conflicts were resolved through discussion between the two authors, and where disagreement occurred, a third author (MC) was consulted and made the final decision about inclusion.

## Eligibility criteria

This review included peer-reviewed qualitative research studies and included surveys and questionnaires when there was an analysis of open-text responses. Eligible studies were required to report on people with ABI over the age of 18 who presented with CCD and/or their adult familiar communication partners and report on the impact of the CCD. Studies were excluded if the person with ABI presented with dysarthria of sufficient severity to preclude them from participating (speech sound disorder), progressive neurological conditions including dementia and terminal brain cancer, myalgic encephalomyelitis or aphasia. Mixed population studies (e.g., aphasia and cognitive-communication changes) were included if the qualitative component was reported separately for each of the diagnoses. Mixed-method studies were included if the qualitative component met the inclusion criteria of the review. Quantitative studies, as well as editorials, conference papers, theses or dissertations and non-peer-reviewed articles from internet websites were excluded.

## Quality appraisal

Included articles were independently assessed by two authors (NB and IC) using the CASP qualitative checklist [39]. Low quality studies were not automatically excluded; rather data from these studies were considered and presented with clear consideration of their quality. The GRADE-CERQual tool [40] was used to assess the degree of confidence in the findings of the qualitative evidence synthesis. The overall assessment of confidence (high, moderate, low, very low) was based on an assessment of four components: methodological limitations, coherence, adequacy, and relevance. The judgement was made independently then discussed to reach consensus agreement between the same two authors (NB and IC).

## Data extraction and synthesis

A data extraction tool was developed, and the first author (NB) extracted the key participant, intervention and methodological information. For the qualitative synthesis, all content under the results/findings sections of the included papers were imported into NVivo 14 (Version 14.23.1) for analysis. The strategy for data synthesis was guided by the methods for thematic synthesis outlined by Thomas and Harden [29]. First, the first author completed free line-by-line coding, whereby every sentence was interpreted and inductively applied to at least one code. New codes were developed when necessary. Second, the free codes were organised into descriptive themes, where similarities and differences between the codes emerged and were grouped into a hierarchical tree structure comprised of themes and subthemes. The first two steps were completed for one of the included articles and reviewed by another author (IC) before the remaining articles were coded in a similar way. Finally, once all included articles were coded and organised, analytical themes were developed using a mind mapping approach which produced an interpretation of the data that went beyond the original studies. Members of the core team independently reviewed the preliminary themes, subthemes and analytical framework and discussed the addition or revision of themes. These themes were also discussed with a patient and public involvement group comprised of people with ABI and their family members, and a clinical reference group of academic and clinical advisors. The final analysis was then reviewed by all members of the team.

## Results

The article selection process is outlined in the PRISMA flow diagram in Fig 1. A total of 4,955 records were identified. After removal of duplicates, 4,050 records were left. Following review of titles and abstracts, 144 papers were identified for potential inclusion. The full texts of these were read and assessed for eligibility (see S3 File for excluded studies with

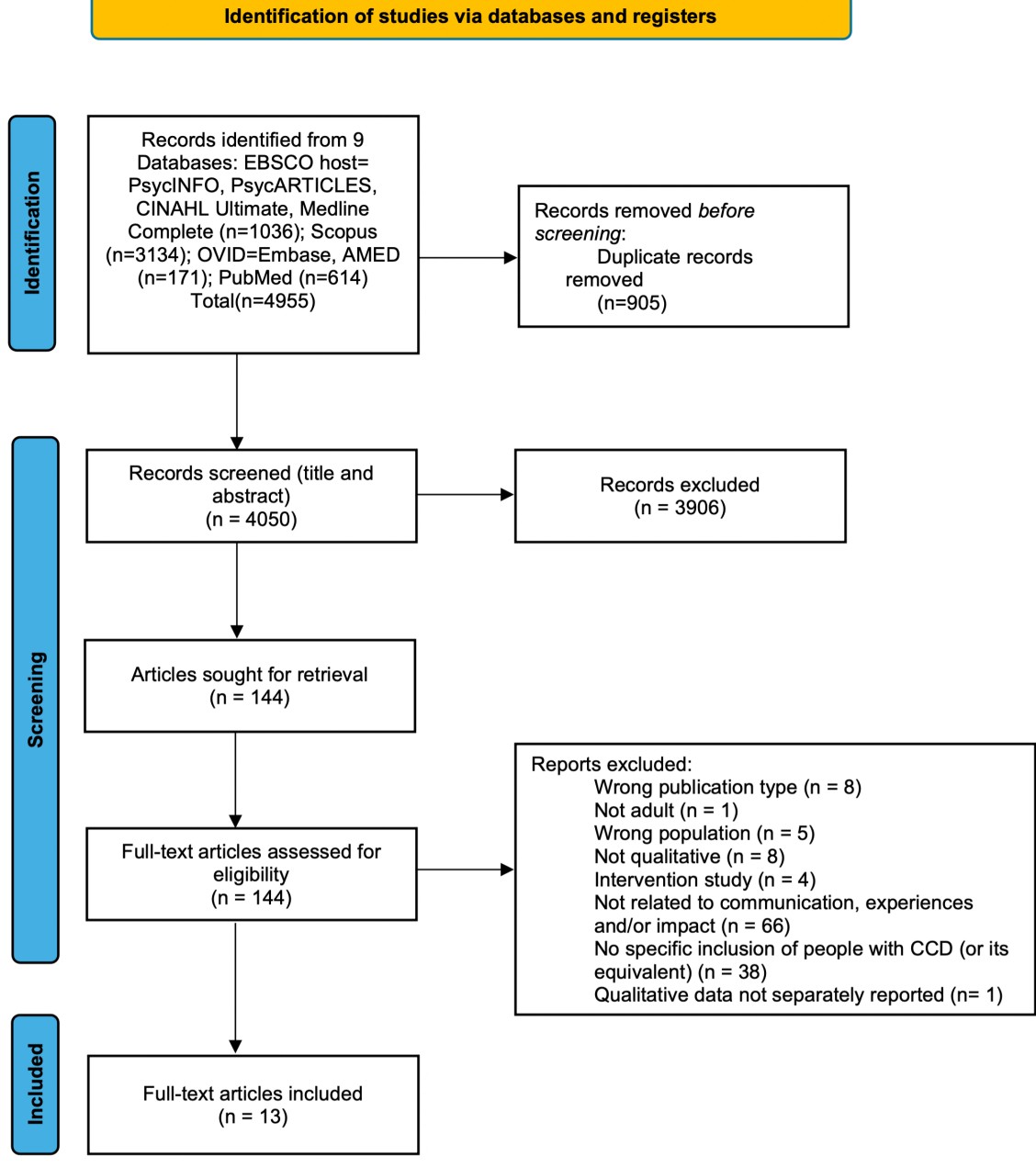

**Fig 1. PRISMA flow diagram.**

reasons for exclusion). Of these, 13 papers met the inclusion criteria and were included in the review. The characteristics of these 13 papers are presented in Table 2.

In terms of the Rayyan mining algorithm used for the initial search, 2315 articles were either labelled as *likely* or *most likely to exclude* at the title and abstract stage of screening. Of these, 1685 were labelled as *most likely to exclude* and these were accurately identified by the text mining algorithm. There were an additional 630 identified as *likely to exclude* and two included papers were identified here. Of these two, one was a minor concern [41] and one a serious concern [42]. A serious concern was identified as a paper that contributed a large proportion of the data (more than 50% of the results) that was used in the synthesis.

## Characteristics of the included studies

The characteristics of the included studies are in Table 2. Most studies (n = 9) were conducted in Australia [43–51] with one study in the United Kingdom [42], United States [41], Netherlands [52] and New Zealand [53]. Across all 13 studies, participants comprised 103 people with ABI and 66 familiar communication partners. There was a broad range of communication partners including spouses, parents, friends, carers, siblings and children.

Most studies (n = 10) involved individual interviews [43–46,48–53]. The remaining studies used focus group methodology [42], online survey [41] and spoken discourse samples [47]. The latter was included as both reviewers could independently identify qualitative information on the participant experience of CCD. A range of methods and theoretical perspectives were used to explore and analyse the data including content or thematic analysis, phenomenology or grounded theory.

More than half of the studies (n = 6) included people with TBI only [41,43,45–47,50]. Two studies comprised both people with TBI and family members [48,49] while a further two studies comprised both people with ABI (including stroke and encephalopathy) and family members [51,52]. One study included friends of people with TBI [44] and two studies included family members of people with TBI [42,53].

As the review was focused on cognitive-communication changes, we documented how studies diagnosed CCD or assessed for the presence of communication changes. In one study [48] a screen of reported ongoing communication difficulties using the cognitive communication checklist for acquired brain injury [54] was used. Two studies [47,52] used formal assessment to identify participants with cognitive-communication changes including use of the La Trobe Communication Questionnaire (LCQ) [55], Functional Assessment of Verbal Reasoning and Executive Strategies [56], Emotion Recognition Test [57] and Faux Pas test [58]. Two studies reviewed the medical files for a documented diagnosis of CCD [42,43]. Bertram and colleagues [44] used an initial sample of discourse rated using the Pragmatic Protocol [59] to determine the presence of CCD, while three studies included participants who self-identified as having CCD [45,51,53]. Four studies did not specify how the CCD diagnosis or communication changes were determined [41,46,49,50].

Most studies (n = 10) were judged to be of high methodological quality on the CASP checklist with no or very minor concerns [42–45,47,48,50–53]. Three studies were judged to raise minor concerns [41,46,49]. Table 3 shows the scores for each study against the CASP checklist. The one item that was either unclear or not discussed in a large proportion of studies (n = 10) related to the relationship between researcher and participant (item 6) and the extent with which the researcher had examined their own role, bias and influence during the study.

## Thematic synthesis

The synthesis explored the experiences of CCDs after brain injury for both people with ABI and familiar communication partners and two main themes and eight sub-themes emerged (Fig 2). There was high confidence in the review findings for seven of the eight sub-themes. There was moderate confidence in the findings of one sub-theme (Communication partner: Relationship and life role changes) which was downgraded due to minor concerns with methodological limitations and adequacy of rich data. See S4 File for all GRADE-CERQual ratings.

**Table 2. Characteristics of included studies.**

| Study name and year, Country | Participant details | | | | | Methodology details | | | |
|---|---|---|---|---|---|---|---|---|---|
| | Person with ABI Type of injury (sample size) | Gender, Age Mean±SD (range) | Severity, TPO Mean±SD (range) | Family member (sample size) | Gender, Age Mean ± SD (range) | Study design | Aim | Data collection | Data analysis |
| Armstrong et al., 2019, Australia | TBI (n=1) | Male Age=41y | Not specified, TPO=10y | NA | NA | Case study | To highlight real-life consequences and issues faced by Aboriginal male brain injury survivors | Semi-structured interviews | Inductive analysis for common themes presented as individual narratives |
| Bertram et al., 2021, Australia | TBI (n=4)[a] | 3 male, 1 female, Age=45 (23-63y) | Not specified, TPO=1-6y | Friends (n=9) | 5 male, 4 female Age=22-90y | Qualitative | To understand processes in the maintenance and development of friendships after TBI | Semi-structured interviews | Grounded theory and constant comparison |
| Brunner et al., 2019, Australia | TBI (n=13) | 7 male, 6 female Age=33 (20-72y) | Moderate-to-severe, TPO=10 (1-59y) | NA | NA | Qualitative | To determine the needs and experiences of people with TBI on their use of social media | Interviews | Content thematic analysis and constant comparison |
| Brunner et al., 2020, Australia | TBI (n=6) | 2 male, 4 female Age=40 (26-72y) | Not specified, TPO=18 (2-59y) | NA | NA | Mixed methods | To examine the experiences and view of people with TBI on their use of Twitter | Interviews | Narrative analysis using realist methods |
| Elbourn et al., 2022, Australia | TBI (n=12) | 11 male, 1 female Age=23-54y | Severe, TPO=6mo-2y | NA | NA | Qualitative | To examine the perspectives of people with TBI towards their communication, recovery and illness narratives | Spoken discourse samples | Reflexive thematic analysis |
| Grayson et al., 2021, United Kingdom | TBI (n=15)[a] | 13 male, 2 female Age=49 (24–63) | Not specified, TPO=4.5 (5m-10.7y) | Parent (n=4), spouse (n=6), sibling (n=3), child (n=2) | 3 male, 12 female Age=51 (19-71y) | Qualitative focus group | To develop a greater understanding of the impact CCD has on family members over time | Focus group | Thematic analysis |
| Kelly et al., 2022, Australia | TBI (n=16) | 5 male, 11 female Age=43.0y (26-70y) | Not specified, TPO=6.3y±3.3 (1-10.7y) | Parent (n=4), spouse (n=7), friend (n=1) | 3 male, 9 female Age=49 (30-72y) | Qualitative | To identify the long-term impacts of CCD as reported by people with TBI and their significant others | Semi-structured interviews | Phenomenology and reflexive thematic analysis |
| Norman et al., 2023, United States | TBI (n=30) | 13 male, 17 female Age=25.5y (18-50y) | Mild, TPO=4.7 (0.9-11y) | NA | NA | Cross-sectional survey | To explore self-perception of CCD of people living with mild TBI | Online survey | Content analysis |

*(Continued)*

**Table 2.** (Continued)

| Study name and year, Country | Participant details | | | | | Methodology details | | | |
|---|---|---|---|---|---|---|---|---|---|
| | Person with ABI Type of injury (sample size) | Gender, Age Mean±SD (range) | Severity, TPO Mean±SD (range) | Family member (sample size) | Gender, Age Mean ± SD (range) | Study design | Aim | Data collection | Data analysis |
| O'Flaherty et al., 1997, Australia | TBI (n=5) | 3 male, 2 female Age=37.6±10.9y (27–52) | Severe, TPO=6.9±6.7 (2.8-18.8y) | Parent (n=1), spouse (n=4) | 1 male, 4 female Age=43±11.8 (31-59y) | Qualitative | To explore the experience of chronic cognitive-communication difficulties following severe TBI | Semi-structured interview | Thematic analysis |
| Shorland & Douglas, 2010, Australia | TBI (n=2) | 1 male, 1 female Age=22.5, 30y | Severe, TPO=15y, 2y | NA | NA | Qualitative | To describe how two adults living in the community with severe TBI construct meaning about their communication and its impact upon friendships | Interviews | Grounded theory |
| Skromanis et al., 2025, Australia | 4 stroke, 5 TBI (n=9) | 5 male, 4 female Age=50.8±12.7 (22–62) | Not specified, TPO=17.0±11.2y (0.7-33.0y) | Carers (n=5) | 5 female, Age=32.6±9.3y (24-48y) | Qualitative | To understand how individuals with ABI experience social disinhibition | Interviews | Thematic analysis with semantic coding approach |
| van den Broek et al., 2025, Netherlands | 6 stroke, 2 TBI, 1 posta-noxic encephalopathy (n=9) | 7 male, 2 female Age=54.7±10.4y (35-67y) | Not specified, TPO=2.9y±0.5y (2.5-4y) | Partners (n=9) | 2 male, 7 female Age=53.4±11.2y (34-66y) | Qualitative | To examine experiences of social cognition problems on relationships | Semi-structured Interviews | Thematic analysis |
| Van-Solkema et al.,2025, New Zealand | 12 TBI (11 male, 1 female)[a] | NA | Not specified, TPO=6.3y (1-36y) | Parent (n=4); Spouse (n=6); Child (n=1) | 1 male, 11 female | Qualitative | To understand and explore families' experience of attention-related communication difficulties following TBI | Semi-structured interviews | Reflexive thematic analysis |

*Note.* TPO=time post-onset; m=months; y=years; NA=not applicable

[a]People with TBI were not active participants in the study but linked to communication partners who were the participants to be involved.

### Person with brain injury

**Communicating is not easy.** Participants with ABI highlighted that communication was not easy. They described trouble understanding, difficulties in their ability to listen and process what had been said, and then to retrieve, structure and organise their responses clearly and coherently, using the right words:

*Sometimes I feel I struggle to find words while talking but more because I do not know how to describe the situation or my feelings not so much because I am struggling to speak* (ABI, p893) [41]

Word finding difficulties led a person to take more frequent pauses to think of the word they wanted to say, to speak around the target word or pause to correct errors in chosen words. Impaired cognition (e.g., memory, concentration

Table 3. CASP ratings.

| Criteria | Armstrong et al., 2019 | Bertram et al., 2021 | Brunner et al., 2019 | Brunner et al., 2020 | Elbourn et al., 2022 | Grayson et al., 2020 | Kelly et al., 2022 | Norman et al., 2023 | OFlaherty & Douglas, 1997 | Shorland & Douglas, 2010 | Skromanis et al., 2025 | van den Broek et al., 2025 | Van-Solkema et al., 2025 |
|---|---|---|---|---|---|---|---|---|---|---|---|---|---|
| Is there clear description of study aims? | Y | Y | Y | Y | Y | Y | Y | Y | Y | Y | Y | Y | Y |
| Is the qualitative methodology appropriate for this study? | Y | Y | Y | Y | Y | Y | Y | Y | Y | Y | Y | Y | Y |
| Was the research design appropriate to address the aims of the research? | Y | Y | CT | Y | Y | Y | Y | N | Y | Y | Y | Y | CT |
| Was the recruitment / sampling strategy appropriate to address the aims of the research? | Y | Y | CT | CT | CT | Y | Y | Y | N | Y | Y | Y | Y |
| Was the data collected in a way that addressed the research issue? | Y | Y | Y | Y | Y | Y | Y | Y | Y | Y | Y | Y | Y |
| Has the relationship between researcher and participants been adequately considered? | N | Y | N | N | N | N | N | N | N | N | Y | N | Y |
| Have ethical issues been taken into consideration? | Y | N | Y | Y | Y | Y | Y | Y | N | Y | Y | Y | Y |
| Was the data analysis sufficiently rigorous? | Y | Y | Y | Y | Y | Y | Y | CT | Y | CT | Y | Y | Y |
| Is there a clear statement of findings? | Y | Y | Y | Y | Y | Y | Y | Y | CT | Y | Y | Y | Y |
| Is the research valuable? | N | Y | Y | Y | Y | Y | Y | Y | Y | Y | Y | CT | Y |
| Total | 8 | 9 | 7 | 8 | 8 | 9 | 9 | 7 | 6 | 8 | 10 | 8 | 9 |

Note. Y=yes; N=no; CT=can't tell.

and processing time) was described as impacting communication. Participants with ABI had difficulty processing multiple pieces of information, difficulty responding quickly and remembering what they wanted to say, and having to repeat themselves:

> *I'd forget about what I was saying* (ABI, p470) [47]

> *I used to be quick off the mark but I've sort of lost all that* (ABI, p891) [49]

One participant with ABI described how these difficulties put them "on the back foot" in conversation [49] with problems starting, maintaining and ending a conversation:

> *I had trouble with continuing a conversation. You say "hi, how are you" and then where do you go from there?* (ABI, p574) [50]

Some participants with ABI reported difficulties in knowing when to take a turn, following multiple conversations in a group situation, or simply keeping up in a fast-moving conversation in a noisy environment. Some family members reported that individuals with ABI gave excessive detail, spoke in an incoherent manner, were unable to use language flexibly for different people and contexts, were impulsive, self-centred or egocentric, made inappropriate sexual comments, struggled

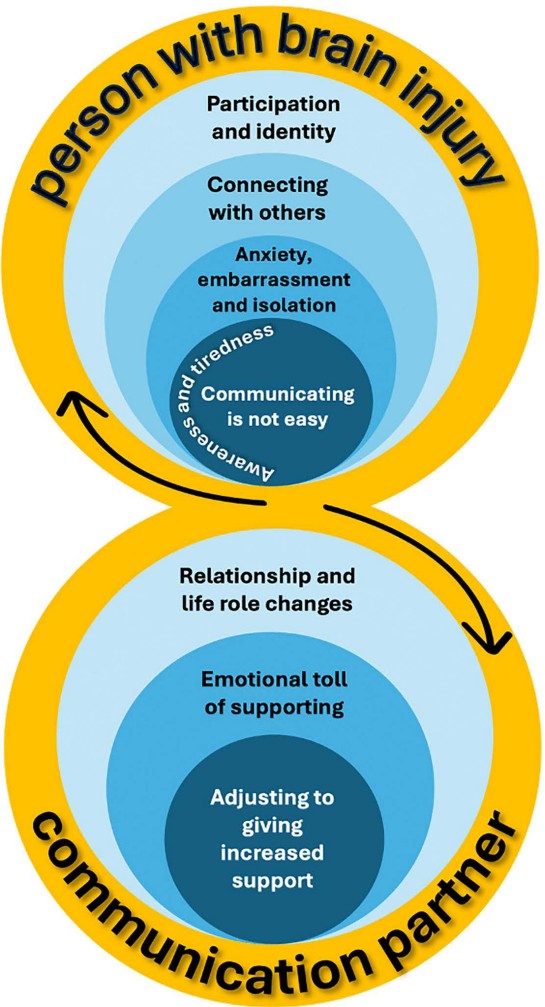

**Fig 2. Themes that emerged for the person with brain injury and the communication partner.**

to understand non-literal language (e.g., irony, jokes) or be sensitive to the subtle uses of language (e.g., being sincere, tactful). Some participants with ABI reported trouble processing the emotions of others and responding in a sincere manner, often responding without thinking:

> *But I also say things I shouldn't say and I don't mean to, I just find, it's almost like I'm thinking out my mouth* (ABI, p6697) [51]

These difficulties were reported by participants with ABI to be ongoing and persistent [48,49]:

> *It took a long time to understand what I now know is cognitive-communication disorder. I'd give too much information in the middle of a conversation and lose my train of thought. I thought that would lessen over time, but it hasn't* (ABI, p2136) [48]

Difficulties such as information overload were reported for conversations involving written communication and communicating online. However, some participants with brain injury reported they could respond to online communication with ease and were able to mitigate some of their cognitive difficulties [46].

**Lack of awareness and feeling tired.** Two factors were found to have a prominent influence on communication. The first, was a lack of awareness which both participants with ABI and family members reported to impact communication. Level of awareness varied, with some participants with ABI reporting no difficulty, or that they were more capable than others perceived them to be. Emergence of awareness was another theme, where participants with ABI reported that their difficulties emerged during therapy or following feedback from family and friends [42,47–52]:

*When I saw my speech pathologist in rehab I actually said to him 'I'm really wasting your time because I'm fine', then I failed every test he gave me. I was so shocked and that's when I thought, 'gee, there's a really big problem here* (ABI, p2137) [48]

*I think it's the same, but, [my friends] were telling me things that they noticed how my communication is different* (ABI, p574) [50]

A lack of awareness was most clear in the reports of carers and family members, who described a range of difficulties with the altered communication styles of participants with ABI [42,44,48,49,51,52].

*Lack of awareness is a huge thing. It can cause little troubles with your everyday interactions* (Wife, p2137) [48]

*Can't see when the other person is offended or bored or needs to leave. Just doesn't see it. Can't see those emotions* (Family member, p18) [42]

The second factor to influence communication was feeling tired or fatigued which was reported by both participants with ABI and family members to have a frequent impact on communication [41,42,45–48,50–52]:

*By the time it gets to Wednesday he gets exhausted, so he doesn't even have any social skills with me never mind anyone else* (Family member, p18) [42]

Sometimes participants became fatigued from the effort involved to communicate with others and/or from sensory overload [41,46,47,52]. Family members reported a need to manage fatigue with rest periods during the day or remove stimuli (e.g., turn off the television) to have serious conversations. These difficulties were reported to persist many years post injury. People with ABI were described as having a "lithium battery… going to run out of power" that needs re-charging [48].

As a result of the fatigue, people with ABI found communicating difficult. They would show little interest in others, pay less attention during conversation, and listen and talk less due to the increased effort required [42,48,50,52].

*You fatigue quicker because you're managing so much more. I couldn't even have a quick chat. It was too much in my brain, I'd literally want to go to sleep* (ABI, p2137) [48]

One carer reported that lack of sleep could lead to frustration and an increase in socially disinhibited behaviours [51]. If additionally faced with sensory overload, participants with ABI were reported to be more self-focused and/or found it difficult to empathise [52]:

*If you're really tired or you're overstimulated, […] then it also becomes harder to empathize with someone else. Because you're just fully occupied with yourself, so you have little space left to really empathize with someone else* (ABI, p348) [52]

**Anxiety, embarrassment and isolation.** Feelings and emotions were described by many participants to emerge from a person's communication difficulties. These feelings were mainly negative and occurred in the lead-up to a

conversation but also during and after a person's interaction with others. Several participants with ABI reported feeling "flustered" due to difficulties expressing themselves or frustration during conversations. Feelings of anxiety and panic were prominent [41,48,51]:

> *You lose words and you end up saying things that you don't mean or you don't feel like you're explaining yourself properly and then you go away feeling anxious because you're like "what did I say? What did I do?" and then it's just like why bother?* (ABI, p2138) [48]

These difficulties in communication sometimes led participants with ABI to experience feelings of being upset, sadness and concern [41,43,49], and from being perceived by others as someone they are not.

> *He does not leave his community very often as he deliberately tries to avoid people and situations that might trigger his 'short fuse' which developed since the accident. He is acutely aware that people think of him as an 'angry person' and this causes him great sadness and concern, "I get angry at everything all the time. When I don't catch on to what they're actually saying and I get angry. I would like to be judged as a nice person, not a horrible angry person"* (ABI, p128) [43]

Further still, people with ABI developed a deeper sense of embarrassment, shame, fear and insecurity, and reduced feelings of self-worth from their communication difficulties and how they were perceived by others, and a sense of inferiority compared to that of their family member.

> *Sometimes when I'm talking the gibberish comes out, and it involuntary you don't know… I'm just so scared and embarrassed to talk to some people in case what comes out of my mouth isn't good* (ABI, p2137) [48]

> *…she'd [significant other] probably be better able to explain it than me* (ABI, p902) [49]

Combined, these feelings of anxiety, sadness and embarrassment led to avoidance behaviours and feelings of isolation and loneliness from not interacting with others [41,43,48,50,52,53]. Low confidence in being able to interact with others led people to be more passive with friends or to withdraw from social situations.

> *I'm not confident in the words, or finding the words, I, I step back from, yeah I suppose putting myself out there (ABI, p574)* [50]

There were some reports from participants with ABI of positive feelings from communicating through social media. These individuals expressed fun, enjoyment and self-confidence from their written communications which was described as a "way of getting out my frustrations" [46]. One participant described enjoyment from conversations with others:

> *Even though I might have enjoyed it and loved, loved the conversation, I, I don't always let them know, that I've had a great time… if I have to get away quickly or something, I, I'll just say 'I've got to go' sort of and leave without saying, without even thinking to say 'I've had a great time' or 'we, should do this again* (ABI, p574) [50]

**Connecting with others.** Communication difficulties were described by participants to have an impact on the connections and relationships people with brain injury have with family, friends and other members of the community. For many people with ABI, reduced social contacts and loss of relationships were reported [42,48,51] including for a loss of friendships, "my friends have all given up on me" [41]. One person with ABI reported a preference to only interact with family members [43].

Relationships with family members were reported to have changed and be disrupted by changes to communication and behaviour [42,43,48,49,51]. Some families described how changes to a person's personality and employment status "made space for improved relationships with other members of the family" [53]. Although for the most part, connections were more commonly described as tense, volatile and unpredictable at times, characterised by frequent disagreements and anger outbursts.

*I used to yell at my family members, my children, my wife, particularly my wife. I used to yell at the kids, I used to yell at everyone. I used to get so frustrated I went into the garden and just started yelling and screaming at the top of my voice and a neighbour, the woman next door, rang up and complained* (ABI, p6696) [51]

*Anything can trigger it. A hair could trigger it. It's like living on a volcano. I never know when she's gonna blow* (Significant other, p903) [49]

Some of the tension within relationships resulted from conversations simply not being the same [42,49–51].

*…some of the problems with friends… part of the friendship was always banter. Banter that you used to… and you [directed at injured partner] can't keep up with that now…You know jokes and asides'* (Significant other, p903) [49]

These communication difficulties, many of which were impacted by lack of awareness, meant that participants with ABI were negatively perceived by others as rude, impolite, and offensive, and lacking in empathy and failing to display tact or to read the emotional cues of others, and not considering the thoughts and opinions of others [42,50,51].

*He's very self-involved. He really does just think about where he's going, what's happening to him. He doesn't really think about me or take other people into consideration* (Family member, p18) [42]

Participants with ABI expressed a desire to form and maintain connections with others, however doing this was clearly problematic [50]. Some participants with ABI described an increased reliance on family for support [49], others felt worthless or inferior, unable to keep up and deal with the pressures of social connection, and so they withdrew or avoided contact with others [42,49,50]. Others simply persisted, despite reduced social contacts [50].

*He doesn't really want to engage in conversation with anybody so you can imagine come Friday when I'm trying to get him to meet up with other people, it doesn't work* (Family member, p19) [42]

Some described friendships as positively maintained despite communication difficulties [44]. Friends reported that they provided advice, shared stories and socialised. Friends perceived a change in the skills of people with ABI and recognised a need to modify their own behaviour and skills to maintain normality. To do this, friends would set boundaries, give prompts, feedback and reassurance to manage the friendship and often advice-giving was described to have been given more by the friends than received.

*You had to learn to talk to him again, instead of just talking to whoever else was in the room about him* (Friend, p87) [44]

Furthermore, social media provided a way for participants with ABI to connect with family, friends and others [45,46]. Participants with ABI could be more active (e.g., writing a post, liking, sharing) or less active, acting more like an observer (e.g., reading posts). They developed awareness of online bullying and harassment and learnt ways to manage these exchanges (e.g., avoid divisive topics, ignore or block people). Active engagement gave people with ABI the opportunity

to maintain or build new relationships with others and led to a sense of belonging from connecting with others. They had the opportunity to think about topics to talk about and took the time to respond thus compensating for their cognitive difficulties.

*I use Facebook to communicate with people, like overseas and stuff, abroad… Even close to home, like reconnecting with other friends* (ABI, p227) [46]

**Participation and identity.** Participants described how cognitive-communication changes impacted the activities people engaged in and their participation with others [48,49,51,52]. Concerns included socially disinhibited behaviours, talking to a group of people, receiving or giving complex information, following and responding to a fast-moving conversation, and making jokes.

*…now she says really inappropriate, offensive things, Um, there's just no filter there whatsoever…usually they would be verbal comments. For example, we'd be going out to the supermarket, um, to get a couple of items and she might see someone who's quite overweight at the shops and point, actually point like this, and go 'oh my God, they're so fat, why are they buying chocolate? They don't need that' or you know, something that's quite inappropriate and you're like oh God, like you can't say that. You know, um, so she does things like that often* (Carer, p6697) [51]

These difficulties led to reduced social participation. People with ABI participated in fewer social activities compared to their pre-injury lives [42,44,48,49,53]. They reported fewer activities with family, were reluctant to attend social events and be a hindrance to those close to them [49,53] and required prompts from family to engage with others [42]:

*She's still not got into phoning up her friends, so I have to say go and phone, go phone* (Family member, p18) [42]

Nevertheless, participation was still reported in some activities including exercise, meeting for coffee, sharing a meal [44] and engaging online to find a date [46] however, doing the activities as they once did was reported to be problematic. Early post-injury, linking therapy goals with social activities was described (e.g., practicing skills by reading aloud to the children each night) [47].
Participants described negative changes to identity and sense of self post-injury, with aspects of the pre-injury self-becoming lost or altered [42,43,48–50,53]. Life roles were described as changed such as being a partner, parent or work colleague. For example, marital relationships were described more like friendships, individuals were less likely to take on parental roles and participants with ABI lost their identity following not returning to work.

*I literally did not know the person that I was anymore. I couldn't communicate properly, I couldn't parent anymore. I couldn't function in this world properly* (ABI, p2138) [48]

*My son doesn't pick up the guitar because Dad's not doing it with him anymore. Because that was their thing. They would chat and play together* (Family member, p19) [42]

Changed communication skills disempowered participants with ABI, with a loss of identity of being good communicators and being well-liked and active, rather than passive in conversations with others [43,49,50]. Social media in part, provided an opportunity for participants with brain injury to establish new identities or "personas" online [46]. Finally, participants with ABI lost their independence, struggling to return to education or employment [41,43,48,49,51,53]. Socially disinhibited behaviours and low frustration tolerance were reported to make return to work difficult. Communicating in a pressurised, busy and distracting workplace and/or education environment was particularly challenging, where receiving or giving complex information, responses and rapid problem-solving was required. Reports of successful return to work involved strong support networks, opportunities for volunteer work and a focus on single (rather than multiple) tasks at once [49,51,53].

 

## Communication partner

**Adjusting to giving increased support.** Family members, friends and carers described the increased physical, financial, and emotional support given to manage the changes in communication for individuals with ABI. They described themselves as people who learn to tolerate and manage the different situations that frequently arise. They reported learning methods to interact differently that included setting boundaries, walking away from arguments, not reacting to anger, redirecting, or refocusing attention, providing memory reminders (e.g., to call friends), avoiding certain topics, summarising conversations, providing reassurance, giving key words and clarifying what someone said, and explaining social inappropriateness. They had to make sure someone was physically available to observe and intervene if needed.

*We really need to make sure that somebody's with him to support him… as he'll come back and tell you things. And it's not accurate either* (Family member, p18) [42]

Family members, friends and carers described the complexity of needing to adjust and change to this increased support for individuals with ABI to achieve success. This required them to adapt and be flexible for individuals with ABI whose communication styles can be somewhat unpredictable. They learned to anticipate what the individual with ABI may do and provide adequate support; or anticipate what they were saying in a conversation and help them respond appropriately. They needed to use methods to ensure conversations flowed smoothly if the individual was distracted or struggling. In the early stages of recovery, some family members described modifying their own behaviour to successfully participate in social activities together with the individual with ABI. Many reported that they would provide guidance and support, to increase an individual's awareness and help them to understand what behaviours may or may not be socially appropriate.

*So, she needs like that little prompt to, um, like externally, so from, yeah, either my mum or somebody else, to then say like "oh, like that's, it's, we don't really ever say that, that's an inappropriate things to say* (Carer, p6698) [51]

Other communication partners described confusion and uncertainty with knowing what the right feedback was and when best to intervene while respecting an individual's independence and ensuring an equal relationship. They reported a need for information, guidance, and support for managing the different challenges they faced each day. Family members reported difficulty in how to handle different situations and people, and the conflict that may arise if not handled correctly.

*It can be hard because sometimes what she says just doesn't make sense, but she's not aware of it. When it happens, I struggle to know how to handle it. I don't want to tell her what to say, but she doesn't realise what she's said doesn't make sense* (Wife, p2138) [48]

**Emotional toll of supporting.** Giving support to people with ABI was described as taking an emotional toll on family members. Family members reported that they sometimes struggled to cope, with feelings of being overwhelmed and exhausted from being the constant source of support unable to relax, and worn down from being shouted or screamed at, "I feel like a punching bag" [42].

*He's stay with me, but he goes to his partner's sometimes and that's been a godsend, because I think if he was 24/7, I would have… I don't know how I would have coped* (Family member, p18) [42]

*I always have to watch the faces of the people he is talking to. To find out if I need to intervene. Can't ever relax* (Family member, p18) [42]

*…I'm struggling to keep my head above water…* (Wife, p7) [53]

Family members reported feelings of frustration and anxiety from having to back down from arguments and providing support all the time as the individual with ABI was not aware of their communication difficulties. One family member reported that they "shut down after a while" [53]. They also described feelings of disappointment and distress caused by the person with ABI being unable to understand their emotional needs and those of others and respond flexibly to feelings and feedback.

> *It's all about how he's feeling which I find very difficult because sometimes I think it would [be] nice if he could give me a hug and ask me how I'm feeling but that doesn't happen* (Family member, p18) [42]

> *I was telling a story but I got no reaction: no question, no…nothing. Am I am quite chatty but at a certain point […] you think: never mind […] It made me feel disappointed that he did not react in a nice, sociable way. […] I was just disappointed* (Partner, p344) [52]

Family members expressed feelings of loss, sadness, loneliness and isolation arising from losing their partner and confidante. Feelings of sadness were expressed from loss of a partner; and loneliness and isolation by several family members who described the lack of someone to talk to or confide in, having to withhold their own thoughts and feelings as they would not be shared by the individual with ABI.

> *I still have no one to confide in, to talk to, you know? Most couples talk about money and finances and everyday things. I feel like I'm on my own a lot* (Wife, p2138) [48]

**Relationship and life role changes.** The changes to communication had a significant impact on relationships between people with brain injury and their family members. Relationships were described as lost, disrupted and less cohesive. There was a reported shift in the dynamics from a collaborative and equal relationship to one defined as solitary or dependent. Family members needed to apologise to others and manage negative views from others including from within the same family where there was a sense of denial and lack of family cohesion. The impact was more apparent for spouses than parents and siblings, "we used to be such a good team" [42]

> *We have never considered leaving each other before, but in the past few years I have questioned whether this is what I want* (Partner, p346) [52]

Despite these negative changes, some participants with ABI described more positive interactions from a more conscious awareness of their own and their partner's thoughts and feelings and therapeutic help for impaired social cognition. Individuals with ABI were described as more open and paying attention their feelings and their partners:

> *He has become much more open, much softer […] just talking a lot more about feelings, expressing things, a lot earlier* (Partner, p349) [52]

As the relationship between person with brain injury and family member changed, so did the life roles. Some of these evolved from a sense of co-dependence. Family members perceived themselves in the role of carer or support worker, whereby they would tell the person with ABI what to do and/or provide support to help.

> *He does rely on me even the simplest ordering in a restaurant, 'do I like this? We've been here before haven't we?' I say 'yes you have and you ordered this' that sort of co-dependence* (Significant other, p2138) [48]

Other family members described how their spousal relationship was one of friendship and that roles and tasks their loved ones would have done before the injury had changed.

*But when you're talking about the relationship with him, we're very close but we don't talk like husband and wife any more, it's more like I'm his pal* (Family member, p19) [42]

*Our roles changed, I mean D.E. used to pay the bills and do all that sort of thing. And all that changed because I had to take over* (Significant other, p901) [49]

## Discussion

The current synthesis explored the experiences of cognitive-communication changes from the perspectives of individuals with ABI and familiar communication partners. The review highlighted the complex and pervasive nature of these changes and how factors such as fatigue and impaired awareness can undermine the most basic of interactions with others. These changes shape how someone feels, can place undue stress and strain on friendships and relationships and disrupt every-day activities with others. Crucially, these changes are not experienced in isolation and are equally felt by those around the individual with ABI. Familiar partners experience struggle and burden as they adapt to altered communication styles and increased caregiving demands. This takes a significant emotional toll on partners as they navigate how best to care and support an individual with ABI.

Participants described a diverse range of cognitive-communication changes that impacted the ability of an individual with ABI to successfully converse with others. Responding quickly and coherently, integrating multiple pieces of information, remembering what to say, reading social cues, generating topics and keeping up in a fast-moving conversation were all examples of communication difficulties reported by participants with ABI. These difficulties highlight how subtle changes in communication can considerably disrupt their everyday conversations. Many participants were also many years post-injury which is consistent with previous findings that CCD is ongoing and pervasive [14,15,48,60]. They not only hinder successful communication but impact emotions, relationships and broader social participation and sense of self, for *both* the individual with ABI and their familiar communication partner.

Lack of awareness into one's own cognitive-communication changes was a factor that influenced successful communication. Some participants with ABI described emergent awareness following feedback from a therapist, family or friends, however; reports also came from family members who identified problems with everyday interactions where individuals with ABI had no awareness of how they conversed with others. Family members described disruptive changes to conversation including speaking in an offensive, unfiltered and inappropriate manner, unable to read social cues or consider the thoughts and opinions of others. These suggest significant changes to social cognition skills, which involve the ability to process emotion, make social inferences and respond appropriately to social cues [61]. Such skills are infrequently or never assessed by rehabilitation professionals [62]. Lack of awareness into cognitive-communication changes and a tendency to overestimate one's abilities is not new [13,63]. However, rehabilitation interventions should address awareness particularly, of cognitive-communication changes [64–66]. The presence of awareness (or lack thereof) has been shown to be integral for motivation and engagement and success in rehabilitation [67,68].

A second factor described by all participants to have exacerbated a person's cognitive-communication changes was fatigue. The influence of fatigue in brain injury is well documented [69–71] and many participants with brain injury in this review recognised that fatigue led to less participation in conversations. Rehabilitation professionals should identify fatigue as an important factor that can influence communication and educate others about how it may impact social participation and engagement in rehabilitation, social, recreational, vocational and academic activities [12,72].

Negative feelings and emotions arose from cognitive-communication changes experienced by participants with ABI, confirming previous reports [73,74]. Participants with ABI felt sad or anxious to interact with others, feeling a sense of shame or embarrassment and low confidence as to how others may perceive their communicative competence or lack thereof. These emotions can result in situational avoidance, social isolation and feelings of loneliness, as has been reported elsewhere [26,75]. This is significant as feelings of loneliness predict quality of life and emotional well-being [76];

however, are challenging to support post-injury [75]. In people with ABI, studies have described the importance of a strong therapeutic rapport, early identification of feelings and emotions of concern, education to families and providing opportunities for connection with others to address emotional consequences of brain injury [26,75].

The lack of communicative competence and negative emotions have an impact on a person's connections with others. Some connections were characterised by conflict and tension due to CCD such as being socially inappropriate, impulsive, disinhibited and misreading social cues. This led others to have negative perceptions of people with ABI with disruption and loss of existing relationships and friendships. Subsequently, participants with ABI participated in fewer social activities with loss of independence from failing to return to work or school and loss of identity and sense-of-self from being good communicators pre-injury. Loss of social networks and difficulty forming new friendships and relationships is not uncommon post-injury [77–79] as are negative changes to identity and sense of self [80].

Rehabilitation should provide opportunities for individuals with ABI to build connections and relationships with others through group-based interventions or setting social participation goals for real-life situations [12]. Training partners who they regularly communicate with may assist to (re)build pre-injury friendships and relationships, while also (re)constructing a positive identity [81,82]. Incorporation of outcomes that address emotional health, social participation and well-being alongside communication outcomes will further ensure that interventions address the multidimensional nature of CCD. Crucially however, there needs to be greater public awareness of CCD and the impact these changes can have not only on individuals with ABI but communication partners as well. Through greater public awareness, including for other healthcare professionals [54,83] partners may be afforded greater empathy for the struggles they experience, and fewer breakdowns may occur in conversations involving individuals with CCD.

A key finding of this review is the impact of CCD on family members, carers and friends. Communication partners often bear the brunt of managing the changes in the individual with ABI daily, navigating social situations and taking on the burden of facilitating conversations. Families experience significant emotional stress, burden and reduced well-being from disruption to their own lives, increased caring responsibilities and unhealthy family functioning [84–87]. This is particularly challenging for individuals with ABI, as their social networks are often limited to close family members, placing the burden of caring on fewer people [78]. This review highlighted the emotional toll and burden unique to supporting individuals with ABI with CCD including stress, anxiety, frustration, and feelings of being overwhelmed and exhausted from the frequent need for support. We already know that the strain of supporting someone with CCD is pervasive and that family members want training, education and support [88] Ensuring that family members have adequate social support systems for themselves is vitally important [89,90].

Family members are trying to support the communication skills of individuals with brain injury with unmet needs for education or support. Families have reported education and training in helpful strategies to support conversations, relationships and social activities as one of their most important needs [90]. In this study, individuals with ABI became more dependent on their family members and friends whose life roles changed as family members or friends become carers or support workers, and spouses became more like friends. Relationship dynamics fundamentally change from brain injury. More specifically for individuals with CCD, partners must provide increased support for the person to navigate everyday interactions with others, which may or may not be successful. Where positive relationships were reported, they were in contexts where patience and tolerance were shown. The role of rehabilitation professionals is to support partners to create positive and supportive communicative environments in which successful conversations can occur with little effort or monitoring. This approach promotes confidence while helping others to develop the skills to manage challenging situations [91,92].

## Clinical implications

The findings from this review highlight the importance of CCDs and their influence on emotions, relationships, social participation and identity for *both* the individual with brain injury and family members, friends and carers. Maintaining social

connection and participation should be considered a fundamental goal of rehabilitation [75,93]. However, effective support requires not only rehabilitation focused on the person with ABI but *also* support for family members. Services need to address the needs of family members and caregivers [84] particularly, their emotional needs and ability to cope [94]. A key intervention recommended for individuals with ABI is communication partner training [12,93]. This provides education and training of the skills required by a family member to improve their conversations with the person with brain injury [95]. An integral first step for health professionals would be to consider the most important communication partners within a person's existing social network [78]. This partner may be an ideal candidate for education and training. In addition, training is needed for healthcare professionals about CCDs after brain injury, to help improve their own interactions with individuals with ABI [96] and identify who may require support. Publicly accessible training programmes for family and healthcare professionals such as *Interact-ABI-lity* offer structured guidance to improve understanding of cognitive-communication changes and their impact after brain injury [97]. In addition, familiar communication partners should be offered support for their emotional needs [90,98] which may include access to counselling and psychological services, and peer support networks.

### Limitations and directions for future research

There are several limitations of this study, most notably, fewer studies that explore the experiences of family members, with most focusing on parents and spouses. There were few studies that included people with non-traumatic brain injuries. Most studies originated from Australia which may limit transferability to other countries and cultures. Many of the excluded studies either reported on people with multiple communication diagnoses including aphasia (e.g., for people with brain tumours), or reported on the impact of the injury and communication problems without specific reference to a communication diagnosis. In relation to the use of a text mining algorithm one study of major concern was excluded erroneously on how CCD was previously defined. This highlights the importance of having inclusion criteria clear at the outset. There is an additional challenge in relation to the identification and diagnosis of CCD. Our review demonstrated heterogeneity in how CCD was identified with diagnostic practices ranging from formal assessments to self-report and case note review. A common challenge for clinicians is choosing an assessment measure that has sufficient sensitivity and ecological validity to detect CCD [72]. The inconsistency in diagnosing CCD risks under-recognition, as people with brain injury with intact surface-level language may be mistakenly judged as communicatively competent. Development of a standardised screening tool sensitive to the identification of CCD, analogous to brief aphasia screeners, is therefore imperative. Already, there is preliminary work underway on the development of a mobile health application to screen for CCD after right hemisphere stroke, which may prove promising [99]. Moreover, all studies were drawn from western countries with most originating in Australia. Further studies from culturally diverse populations, including low and middle-income countries is needed to consider the cultural influences of cognitive-communication changes internationally.

While many involved in-depth interviewing and analysis of participants, some studies collected limited qualitative data on the impact of CCDs, which limited the richness of data to analyse. There were minor methodological concerns across most studies as determined by the CASP tool. Most studies did not sufficiently consider the relationship between the research and participants (CASP item 6) which negatively affected the quality ratings. Encouragingly, there was a high degree of confidence in the findings as determined by the GRADE CERQual tool, which lends support to synthesised themes and sub-themes.

Consideration of the impact of CCD is important when thinking about the potential outcome of rehabilitation. This synthesis guides researchers as to the questions they may ask during in-depth interviews considering the wider, broader impact of changes beyond the level of the individual (e.g., activities, participation, well-being).

### Conclusion

By synthesising qualitative studies on CCD, this review sought to explore the lived experience of CCD for both individuals with brain injury and familiar communication partners. Changes to skills in conversation influence broader communication,

emotions, relationships, and identity, with consequences extending to families and friends. Our findings underscore the need for systematic approaches to identify CCD and for interventions that address emotional and relationship changes while providing comprehensive support for communication partners. By embedding improved diagnosis and CCD management within clinical practice, services can more effectively respond to the lived realities of CCD and support individuals with ABI and partners to re-engage in meaningful social worlds.

## Supporting information

**S1 File. PRISMA 2020 Checklist.**
(DOCX)

**S2 File. ENTREQ Checklist.**
(DOCX)

**S3 File. Excluded studies with reasons for exclusion.**
(DOCX)

**S4 File. GRADE-CERQual ratings for themes and sub-themes.**
(DOCX)

## Author contributions

**Conceptualization:** Nicholas Behn, Madeline Cruice, Katerina Hilari, Ian Kellar, Leanne Togher.

**Data curation:** Nicholas Behn, Madeline Cruice, Katerina Hilari, Ian Kellar, Leanne Togher.

**Formal analysis:** Nicholas Behn, Iben Christensen, Madeline Cruice, Katerina Hilari, Ian Kellar, Leanne Togher.

**Funding acquisition:** Nicholas Behn.

**Investigation:** Nicholas Behn.

**Methodology:** Nicholas Behn, Madeline Cruice, Katerina Hilari, Ian Kellar, Leanne Togher.

**Project administration:** Nicholas Behn.

**Resources:** Nicholas Behn, Ian Kellar.

**Supervision:** Nicholas Behn, Madeline Cruice, Katerina Hilari, Ian Kellar, Leanne Togher.

**Validation:** Nicholas Behn, Iben Christensen.

**Visualization:** Nicholas Behn, Iben Christensen, Madeline Cruice, Katerina Hilari, Leanne Togher.

**Writing – original draft:** Nicholas Behn.

**Writing – review & editing:** Nicholas Behn, Iben Christensen, Madeline Cruice, Katerina Hilari, Ian Kellar, Leanne Togher.

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
