## [Decision Letter · Decision Letter 0]

17 Feb 2026

PONE-D-26-01897Lived experience of cognitive-communication changes after acquired brain injury: A qualitative evidence synthesisPLOS One

Dear Dr. Behn,

Thank you for submitting your manuscript to PLOS ONE. After careful consideration, we feel that it has merit but does not fully meet PLOS ONE’s publication criteria as it currently stands. Therefore, we invite you to submit a revised version of the manuscript that addresses the points raised during the review process.

We look forward to receiving your revised manuscript.

Kind regards,

Firas H Kobeissy, PhD

Academic Editor

PLOS One

Journal Requirements:

2. Kindly update the title of your manuscript to include the phrase 'systematic review' in it.

“This study is funded by the NIHR Advanced Fellowship (NIHR302952). The views expressed are those of the author(s) and not necessarily those of the NIHR or the Department of Health and Social Care”

4. We note that your Data Availability Statement is currently as follows: “All relevant data are within the manuscript and its supporting information files”

Reviewers' comments:

Reviewer's Responses to Questions

**Comments to the Author**

1. Is the manuscript technically sound, and do the data support the conclusions?

Reviewer #1: Yes

Reviewer #2: Yes

Reviewer #3: Yes

2. Has the statistical analysis been performed appropriately and rigorously? 

Reviewer #1: Yes

Reviewer #2: N/A

Reviewer #3: Yes

3. Have the authors made all data underlying the findings in their manuscript fully available?

Reviewer #1: Yes

Reviewer #2: Yes

Reviewer #3: Yes

4. Is the manuscript presented in an intelligible fashion and written in standard English?

Reviewer #1: Yes

Reviewer #2: Yes

Reviewer #3: Yes

5. Review Comments to the Author

Reviewer #1: This manuscript presents a well-conducted and methodologically rigorous qualitative evidence synthesis that is clearly reported and grounded in appropriate frameworks (PRISMA, ENTREQ, CASP, and GRADE-CERQual). The search strategy is comprehensive, study selection and appraisal are transparent, and the analytic process is described in sufficient detail to support trustworthiness of the findings. The themes are coherent, well-supported by participant quotations, and the conclusions are appropriately aligned with the data. The inclusion of both people with ABI and familiar communication partners is a particular strength and adds important clinical relevance. I have no major concerns. Minor points the authors may wish to consider include clarifying the ethics statement to explicitly note that ethical approval was not required due to secondary analysis of published data, and briefly acknowledging the predominance of Australian studies and heterogeneity in how cognitive-communication disorder was identified as potential limitations affecting transferability. Overall, this is a high-quality synthesis that makes a valuable contribution to the literature and is suitable for publication with minor revision, if any.

Reviewer #2: This paper examines the lived experiences of cognitive-communication changes following acquired brain injury (ABI) among individuals and their family members. Please see the comments below:

1. I suggest revising the title to better reflect the stated objective, as the study focuses not only on individuals with ABI but also on their family members.

2. The introduction is concise and well written. However, adding more background on sociocultural and ecological factors would strengthen the context, as these factors may significantly influence cognitive-communication changes in this population.

3. In the Methods section, please provide additional detail regarding how studies in different languages were handled during analysis, if applicable. Clarification on the timeline and scope of the literature search would also be helpful.

4. It is somewhat unclear why the inclusion criteria limited the research designs primarily to qualitative and mixed-methods studies. Further justification for this decision would improve clarity.

Overall, this is an engaging and well-written paper, but some areas would benefit from additional clarification.

Reviewer #3: This manuscript presents a qualitative evidence synthesis exploring the lived experiences of cognitive-communication disorder (CCD) following acquired brain injury (ABI), from the perspectives of individuals with ABI and their familiar communication partners. The topic addresses an important and underexplored area within rehabilitation research and offers clinically meaningful insight into the psychosocial and relational impact of CCD. Overall, the manuscript is clearly structured, methodologically sound, and thoughtfully interpreted. The conclusions are well aligned with the findings and are not overstated.

1. Title and Abstract

The title accurately reflects the study design and scope. The abstract is clear, balanced, and appropriately summarizes the methods and findings without overstating conclusions. The aims, approach, and thematic findings are presented concisely and accurately.

2. Introduction

The introduction effectively identifies the knowledge gap related to cognitive-communication changes and clearly justifies the need for deeper understanding of lived experiences. The study aim is explicitly stated and logically follows from the background literature.

3. Methods

The study design is appropriate for the research question. The use of thematic synthesis allows meaningful exploration of participant experience across studies.

The search strategy is transparent, with databases and search terms clearly described. Inclusion and exclusion criteria are well defined.

The use of GRADE-CERQual to assess confidence in qualitative findings strengthens methodological rigor. The description of data extraction and synthesis is clear and reproducible.

4. Results

Results are presented in a logical and systematic manner. The characteristics table of included studies is well organized and non-redundant.

Thematic synthesis is applied appropriately, and the identification of two major themes and eight subthemes appears well supported by the included studies.

5. Discussion

The discussion aligns closely with the findings and appropriately contextualizes them within the broader literature. Clinical implications are thoughtfully considered. Limitations are acknowledged, and directions for future research are clearly articulated.

6. References

The manuscript includes an extensive and up-to-date reference list. Citations appropriately support the claims made throughout the manuscript.

Recommendation

Accept

The manuscript is methodologically sound and contributes meaningful insight into cognitive-communication changes following ABI.

6. PLOS authors have the option to publish the peer review history of their article (what does this mean?). If published, this will include your full peer review and any attached files.

Reviewer #1: **Yes:** Yazan Adam Bouchi

Reviewer #2: No

Reviewer #3: **Yes:** Samed Obeng, MD

---

## [Author Response · Author response to Decision Letter 1]

3 Mar 2026

I have provided a "response to reviewers" document

---

## [Editor Report · Decision Letter 1]

28 Apr 2026

Lived experience of cognitive-communication changes for people with acquired brain injury and familiar communication partners: A qualitative evidence synthesis

PONE-D-26-01897R1

Dear Dr. Behn,

We’re pleased to inform you that your manuscript has been judged scientifically suitable for publication and will be formally accepted for publication once it meets all outstanding technical requirements.

Kind regards,

Firas H Kobeissy, PhD

Academic Editor

PLOS One
---

## [Editor Report · Acceptance letter]

PONE-D-26-01897R1

PLOS One

Dear Dr. Behn,

I'm pleased to inform you that your manuscript has been deemed suitable for publication in PLOS One. Congratulations! Your manuscript is now being handed over to our production team.

Kind regards,

on behalf of

Dr. Firas H Kobeissy

Academic Editor

PLOS One